

# Prognosis value of galectin-3 in patients with dilated cardiomyopathy: a meta-analysis

Yan Xiong[1,2] and  Qing Zhang[1]

[1] Department of Cardiology, West China Hospital, Sichuan University, Chengdu, Sichuan, China
[2] Department of Cardiology, Sichuan Academy of Medical Sciences and Sichuan Provincial People's Hospital, University of Electronic Science and Technology of China, Chengdu, Sichuan, China

## ABSTRACT

**Background.** Accurate prediction and assessment of myocardial fibrosis (MF) and adverse cardiovascular events (MACEs) are crucial in patients with dilated cardiomyopathy (DCM). Several studies indicate that galectin-3 (gal-3) as a promising prognostic predictor in patients with DCM.

**Methods.** A comprehensive search was conducted in PubMed, EMBASE, the Cochrane Library, and Web of Science for relevant studies up to August 2023. The hazard ratios (HRs) of gal-3 for MACEs in DCM patients, and for MACEs in LGE(+) versus LGE(-) groups, were evaluated. Statistical analysis was performed using STATA SE 14.0 software.

**Results.** Seven studies, encompassing 945 patients, met the eligibility criteria. In DCM patients, abnormally elevated gal-3 levels were indicative of an increased MACEs risk (HR = 1.10, 95% CI [1.00–1.21], $I^2 = 65.7\%$, $p = 0.008$). Compared with the LGE(-) group, the level of gal-3 in LGE(+) group was higher (HR = 1.12, 95% CI [1.05–1.19], $I^2 = 31.4\%$, $p = 0.233$), and the combination of gal-3 and LGE significantly improved the prediction of MACEs. Sensitivity analysis confirmed the robustness of all results.

**Conclusions.** This study's findings suggest that elevated gal-3 levels significantly correlate with increased MACE risk in DCM, highlighting its potential as a biomarker. However, significant heterogeneity among studies necessitates further research to ascertain gal-3's predictive and diagnostic value in DCM prognosis, particularly in conjunction with LGE.

**PROSPERO ID.** CRD42023471199.

Corresponding author
Qing Zhang, qzhang2000cn@163.com

# INTRODUCTION

Dilated cardiomyopathies (DCM) represent a significant subtype of non-ischemic cardiomyopathy (NICM). Statistics indicate thatalmost 33% of heart failure cases came from DCM (*Khan et al., 2013*; *Smith et al., 2015*). Clinical studies reveal a close correlation between the heightened living costs and mortality risk in DCM patients and the high incidence of major adverse cardiovascular events (MACEs), including cardiac death, arrhythmic events (ventricular fibrillation and ventricular tachycardia), and exacerbated heart failure (*Mandawat et al., 2021*). Myocardial fibrosis (MF), stemming

from neurohormonal activation and myocardial susceptibility, and associated with the decline in left ventricular (LV) systolic and diastolic functions, emerges as the principal mechanism behind the frequent MACEs (*Bänsch et al., 2002*; *Desai et al., 2004*). Therefore, early and accurate assessment of MF, and prediction the MACEs in patients with DCM, are very helpful for doctors to accurately predict the risk and DCM patients to get timely drug intervention.

Clinically, endocardial biopsy is the most commonly used method for diagnosis and risk assessment of DCM patients, but this method has several limitations that can not be ignored, such as doubtful representativeness of the results caused by small sample size and several complications (*Perazzolo Marra et al., 2014*). Recently, cardiac magnetic resonance imaging (CMR) with late gadolinium enhancement (LGE) has emerged as a preferred diagnostic method among DCM patients. This imaging method can not only accurately identify and quantify ventricular MF (*Mewton et al., 2011*), but also predict some of MACEs of DCM patients by evaluating MF, including hospitalization and death related to heart failure (*Lehrke et al., 2011*; *Masci et al., 2012*; *Wu et al., 2008*). However, given that LGE relies on the contrast in signal intensities between localized MF and healthy myocardium, its capability to identify diffuse interstitial fibrosis in DCM patients is limited (*de Leeuw et al., 2001*), which may lead to misdiagnosis of high-risk patients with NICM. Therefore, it is important to find more potential predictors and study their effects alone or in combination with LGE to evaluate the prognosis of DCM patients.

Galectin-3 (gal-3), a galactoside-binding lectin, is a promising new cardiac biomarker, which was included in the 2013 heart failure management guide and confirmed to have the function of detecting the risk of adverse events (*Yancy et al., 2013*). It has been found that gal-3 is directly related to myocardial collagen turnover, and may be helpful for predicting MACEs (*González et al., 2018*). In addition, the up-regulation of gal-3 in the process of heart disease results in macrophage migration and fibroblast proliferation, which leads to fibrosis (*de Boer et al., 2009*), which has also been confirmed by vergaro who thought galectin-3 may be related to myocardial fibrosis in NICM patients evaluated by LGE (*Vergaro et al., 2015*), but this result still needs a large number of samples to further verify.

In summary, existing evidence suggested gal-3 as a potent prognostic predictor for DCM patients. To further ascertain gal-3's prognostic relevance in DCM, this study conducted a meta-analysis to evaluate: (1) the predictive value of gal-3 alone for MACEs in DCM patients, and (2) the predictive capability of gal-3 combined with LGE for MACEs, compared to LGE alone.

## MATERIAL AND METHODS

The systematic review and meta-analysis adhered to the Preferred Reporting Items for Systematic Reviews and Meta-Analyses (PRISMA 2020) guideline (*Page et al., 2021*).

### Search strategy

A comprehensive search of the PubMed, EMBASE, Cochrane Library, and Web of Science databases was conducted for relevant studies from inception to August 2023. Considering that gal-3 has many former names, we included as many of these terms as possible

in the search strategy for a more comprehensive inclusion of relevant studies. Our searching adopted the medical subject heading (MeSH) such as 'galectin 3', 'CBP-30', 'Galectin-3', 'Dilated Cardiomyopathies', 'Dilated Cardiomyopathy', 'Familial Idiopathic Cardiomyopathies', and 'Familial Idiopathic Cardiomyopathies' and relevant keywords to generate the search strategy. The detailed search strategy for databases is summarized in Table S1.

### Inclusion and exclusion criteria

Inclusion criteria included: (1) patients with Dilated Cardiomyopathy; (2) full text written in English; (3) studies contain at least (or extractable) one of the following outcomes:gal-3 for MACEs in patients with DCM, gal-3 for MACEs in LGE(+) *vs* LGE(-) group;4) study design were prospective studies, retrospective studies, *etc*.

Exclusion criteria included: (1) duplicate reports of the same study; (2) conference abstracts, case reports, and review studies; (3) studies lacking comprehensive data.

### Data extraction

Two independent reviewers (Yan Xiong and Qing Zhang) selected eligible studies, involving title and abstract screening followed by full-text examination. Disagreements between them were resolved through discussions with a third one. Data were collected as previously described by *Jia et al. (2023)*, encompassing details such as author's name, publication year, study design, country, sample size, mean age, and percentage of female patients. Notably, this study specifically extracted information regarding the gal-3 cut-off value and diagnostic criteria.

### Quality assessment

The risk of bias in included studies was assessed by two investigators (Yan Xiong and Qing Zhang) using ROBINS-I. ROBINS-I is a tool for assessing bias in non-randomized studies of interventions (NRSI), available at http://www.riskofbias.info. It addresses various biases, including those related to confounding, participant selection, intervention classification, deviations from intended interventions, missing data, outcome measurement, and the selection of reported results, as well as the overall risk of bias. Based on the assessment results, studies were categorized based on their bias risk as either "Low risk", "Moderate risk", "Serious risk", or "Critical risk".

### Statistical analysis

STATA SE 14.0 software (StataCorp, College Station, Texas, USA) was utilized for the meta-analysis. The hazard ratios (HRs) and 95% confidence intervals (CIs) were used to assess gal-3 for MACEs in patients with DCM, gal-3 for MACEs in LGE(+) *vs* LGE(-) group. Weight mean difference (WMDs) and 95% CIs were selected to compare differences in gal-3 concentrations between patients. Heterogeneity was evaluated using $\chi^2$ and I-squared ($I^2$) tests. The random-effect model was adopted if the $p \leq 0.10$ and $I^2 \geq 50\%$, indicating significant heterogeneity among the studies. Otherwise, the fixed-effect model was applied. Funnel plots, the Begg rank correlation (*Begg & Mazumdar, 1994*) and egger weighted regression (*Egger et al., 1997*) were employed to assess publication bias. In the presence

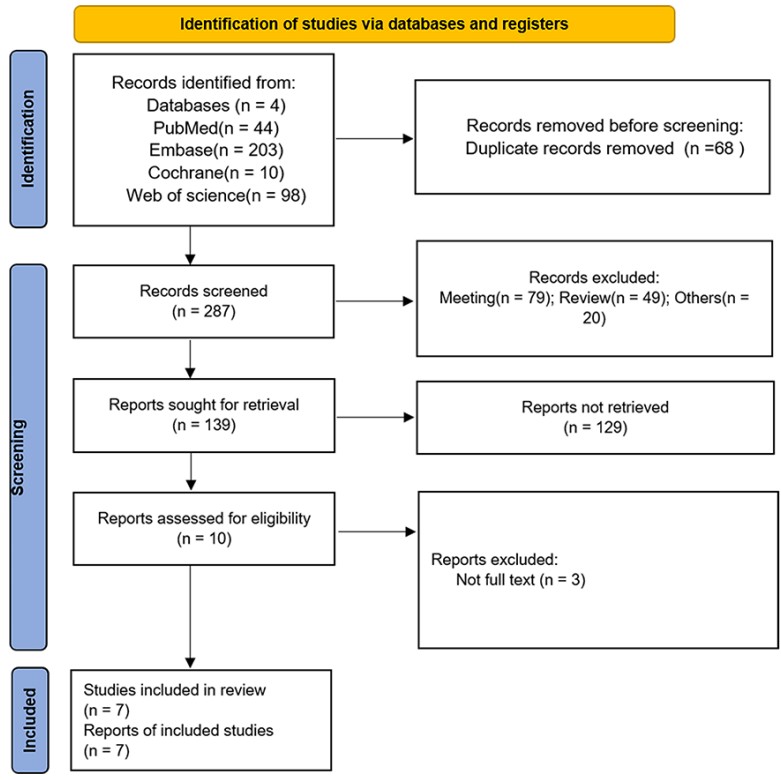

**Figure 1  PRISMA flow chart for study screening and inclusion.**

of significant bias, a trim-and-fill analysis determined the impact of publication bias on the outcomes. Subgroup analysis was used to explore possible sources of heterogeneity. The leave-one-out method for sensitivity analysis tested the robustness of the results. *P* value < 0.05 indicated statistical significance.

# RESULTS

## Study selection

Initially, 355 studies were identified as potentially relevant through database searches. Following the removal of 68 duplicates, 277 records were excluded based on title or abstract review. Ultimately, seven studies (*Binas et al., 2018*; *Hu et al., 2016*; *Karatolios et al., 2018*; *Revnic et al., 2022*; *Rubiś et al., 2021*; *Vergaro et al., 2015*; *Wojciechowska et al., 2017*) were selected for data extraction and meta-analysis after full-text review of 10 manuscripts. The flow chart of the studies was presented in Fig. 1.

## Study characteristics

The seven included studies, published between 2015 and 2021, had sample sizes ranging from 57 to 262. The studies were conducted in one each in Poland, China, Italy, Germany, and Romania. The majority of the study population were middle age. female% ranged from 19.6 to 48.23. The participants' demographic characteristics was shown in Table 1.

**Table 1** Baseline characteristics of seven included studies.

| Study ID | Country | Simple size I/C | Diagnosis | study design | Age (years old) | Gender, female, n (%) | gal-3 cut-off value |
|---|---|---|---|---|---|---|---|
| *Rubiś et al. (2021)* | Poland | 70/20 | The European Society of Cardiology 2007 guidelines | Retrospective study | NA | NA | 18.59 ng/ml |
| *Hu et al. (2016)* | China | 35/50 | The criteria of the American Heart Association | Prospective study | 54.97 | 48.23 | 13.38 u/l |
| *Vergaro et al. (2015)* | Italy | 106/44 | The World Health Organization criteria | Prospective study | 58.53 | 27.33 | 14.4 ng/ml |
| *Wojciechowska et al. (2017)* | Poland | 67/40 | The World Health Organization criteria | Retrospective study | 50.3 | 19.6 | 4.1 ng/ml |
| *Karatolios et al. (2018)* | Germany | 38/19 | The criteria of the position statement from the European Society of Cardiology working group on myocardial and pericardial diseases | Prospective study | 48.9 | 22.81 | 59 ng/ml |
| *Binas et al. (2018)* | Germany | 117/145 | NA | Retrospective study | 50.2 | 24.81 | NA |
| *Revnic et al. (2022)* | Romania | 73/121 | Performed by Cardiac magnetic resonance imaging | Prospective study | 48.7 | 25.77 | 11 ng/ml |

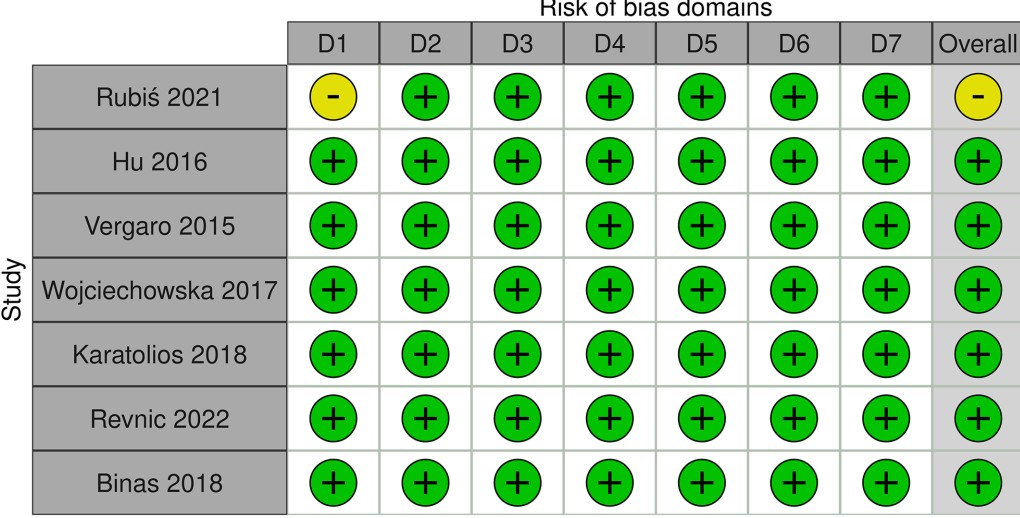

Domains:
D1: Bias due to confounding.
D2: Bias due to selection of participants.
D3: Bias in classification of interventions.
D4: Bias due to deviations from intended interventions.
D5: Bias due to missing data.
D6: Bias in measurement of outcomes.
D7: Bias in selection of the reported result.

Judgement
– Moderate
+ Low

**Figure 2** Risk-of-bias in individual studies using the ROBIS-I.

## Quality assessment

Quality assessment of each included study was conducted using ROBINS-I. Six of the seven studies were judged to have a low risk of bias, and only one study had a moderate risk of bias. The only concern in this moderate-risk biased study is that there may be potential confounding factors (*Rubiś et al., 2021*) (Figs. 2 and 3).

## Gal-3 level for MACEs in patients with DCM

Gal-3 for MACEs in patients with DCM was reported in seven studies (*Binas et al., 2018*; *Hu et al., 2016*; *Karatolios et al., 2018*; *Revnic et al., 2022*; *Rubiś et al., 2021*; *Vergaro et al.,*
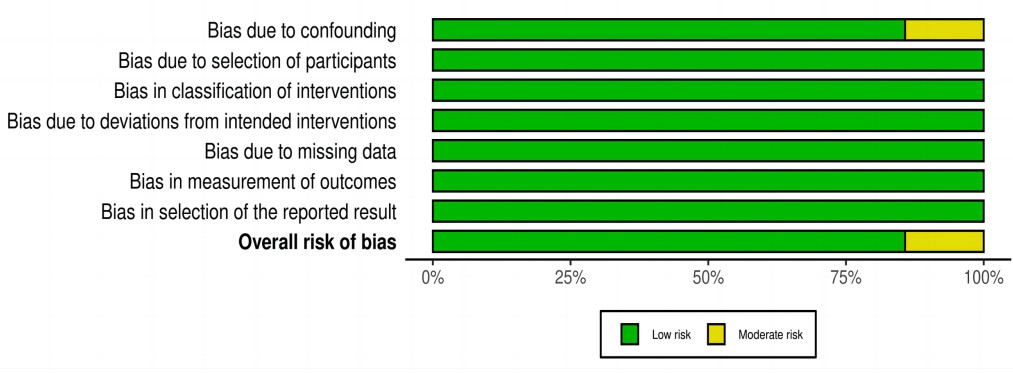

**Figure 3** Risk-of-bias summary using the ROBIS-I.

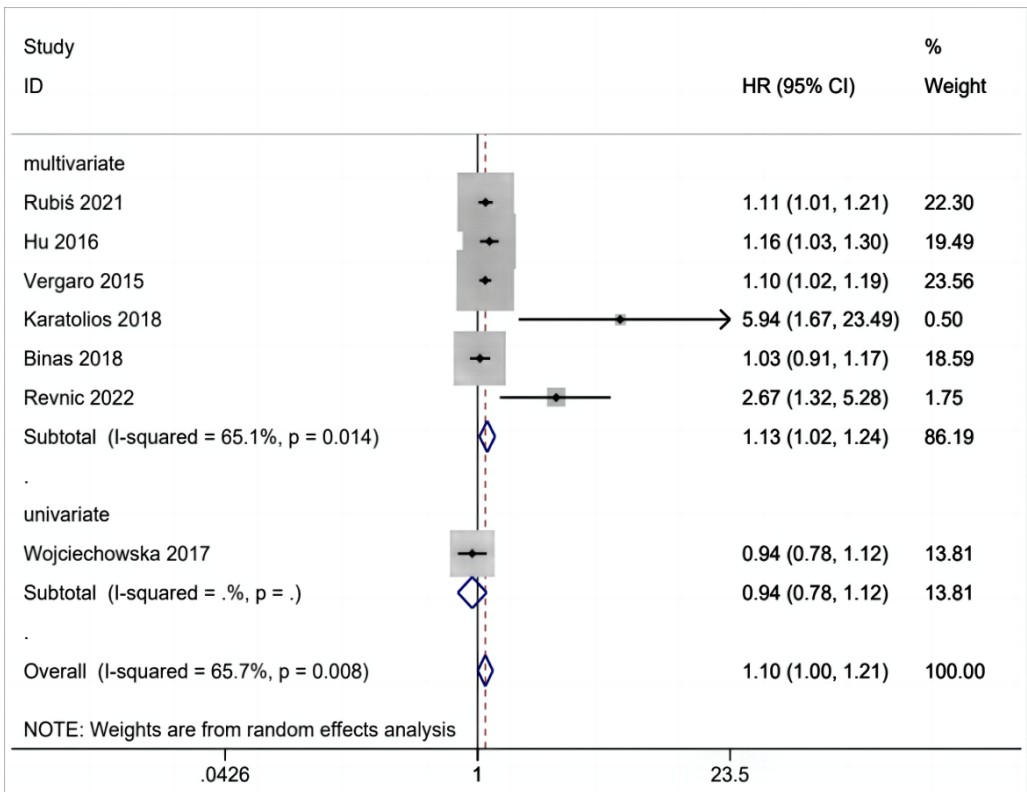

**Figure 4** Forest plot for gal-3 level for MACEs in patients with DCM.

*2015*; *Wojciechowska et al., 2017*) that included 945 patients. From the overall analysis, a greater risk of MACEs was observed for an abnormal elevation of gal-3 level (Fig. 4). The analysis indicated that with significant heterogeneity, the increase of gal-3 level was potentially relevant to a greater risk of MACEs in patients with DCM (HR =1.10, 95% CI [1.00–1.21], $I^2 = 65.7\%$, $p = 0.008$).

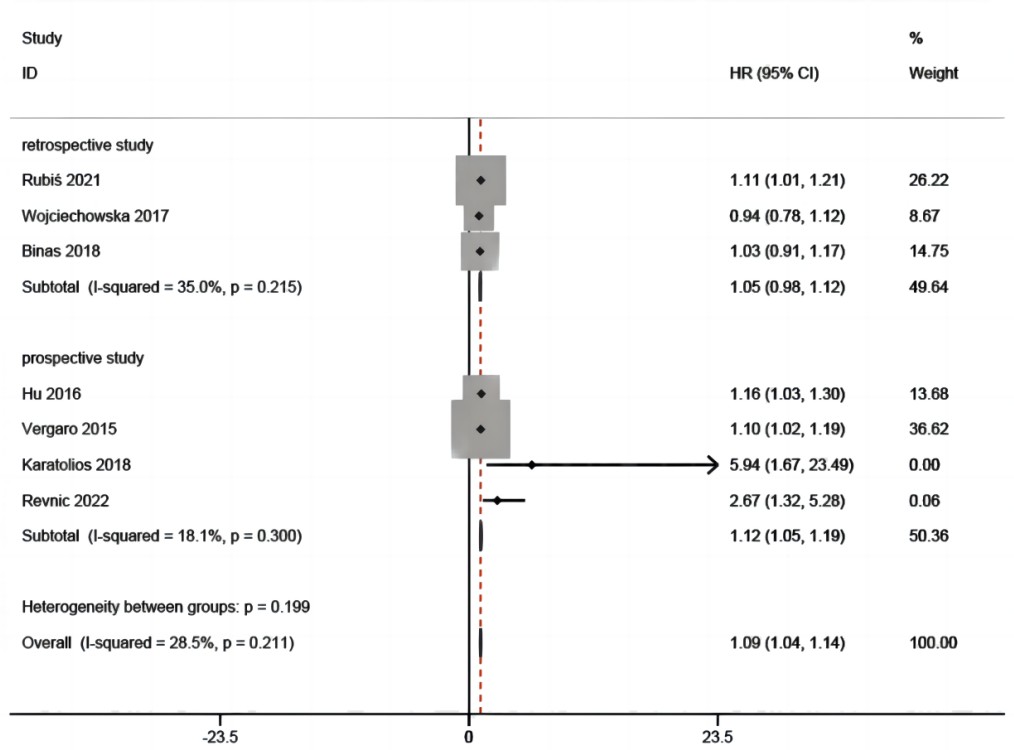

**Figure 5** Forest plot for gal-3 level for MACEs in LGE (+) *vs* LGE (-) group.

## Gal-3 level for MACEs in LGE(+) *vs* LGE(-) group

In included patients with DCM, three studies (*Hu et al., 2016*; *Revnic et al., 2022*; *Vergaro et al., 2015*) were available for the analysis of gal-3 for MACEs in LGE(+) group *versus* LGE(-) group in this meta-analysis (Fig. 5). Compared with LGE(-) group, the level of gal-3 in LGE(+) group was higher(HR = 1.12, 95% CI [1.05–1.19], $I^2 = 31.4\%$, $p = 0.233$),which showed the combination of gal-3 and LGE significantly improved the prediction of MACEs.

## Gal-3 level between different patients

The five included studies provided data on gla-3 levels in patients with different characteristics, including DCM *vs.* healthy individuals ($n = 1$), LGE (+) *vs.* LEG(-) ($n = 2$), LVRR (+) *vs.* LVRR (-) ($n = 1$), and survival *vs.* death ($n = 1$). We hope to indirectly demonstrate the possible predictive effect of gal-3 by analyzing the variation of gal-3 levels among different patients. However, the pooled results showed that the level of gal-3 was not significantly different among patients with different characteristics (HR = 1.20, 95% CI [−3.16–5.56], $I^2 = 93.5\%$, $p = 0.000$).

## Results of subgroup analysis

The above results are analyzed in subgroups by different study designs (Fig. 6) and analysis method (multivariate and univariate) (Fig. 7), and it is found that the conclusion that

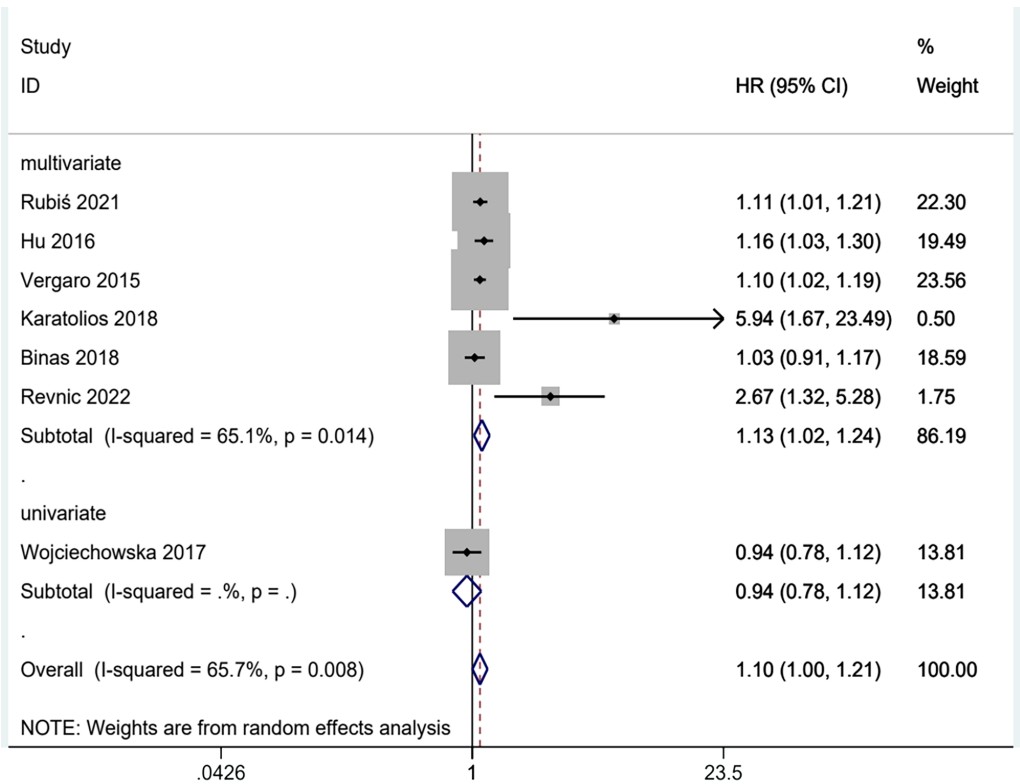

**Figure 6** Subgroups analysis of gal-3 level for MACEs in patients with DCM in prospective and retrospective designs.

the abnormal elevation of gal-3 level indicated a greater risk of MACEs was statistically significant in prospective studies (HR = 1.26, 95% CI [1.02–1.55], $I^2 = 76.3\%$, $p = 0.005$), but not statistically significant in retrospective studies (HR = 1.05, 95% CI [0.96–1.14], $I^2 = 31.4\%$, $p = 0.233$). The results of multivariate group showed that abnormal increase of gal-3 level suggested an increased risk of mace, and the difference was significant (HR = 1.13, 95% CI [1.02–1.24], $I^2 = 65.1\%$, $p = 0.014$). There was only one study in the univariate analysis group, and the strength of evidence was limited (HR = 0.94, 95% CI [0.78–1.12]).

## Publication bias and Sensitivity analysis

The study used the funnel plot, Begg and Egger's test to evaluate the publication bias in this meta-analysis. There may be publication bias in gal-3 in LGE(+) *vs* LGE(-) group (Table S2, Figs. 1 and 2). Therefore, the study used the trim and fill method analysis, and the analysis showed that the bias had little effect on the results of gal-3 in LGE(+) *vs* LGE(-) group (Figs. S3, S4). Sensitivity analysis demonstrated that the pooled effect size results were robust (Figs. S5, S6).

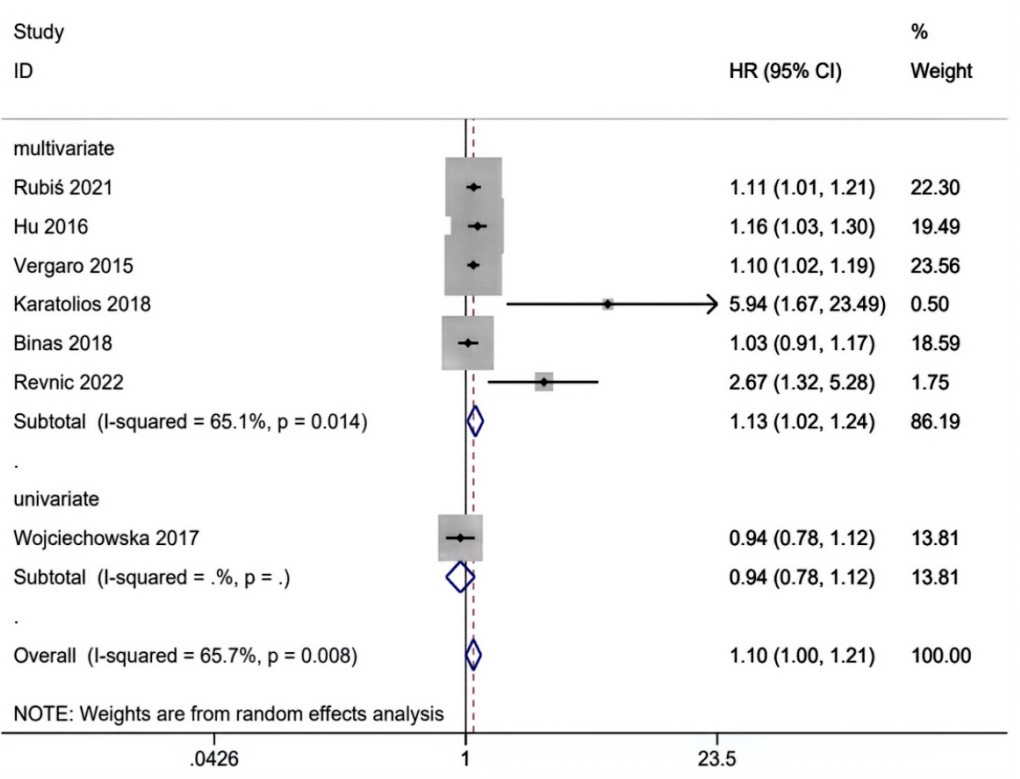

**Figure 7** Subgroups analysis of gal-3 level for MACEs in patients with DCM in multivariate and univariate studies.

## DISCUSSION

This meta-analysis, encompassing seven studies, highlighted significant differences in gal-3 levels among DCM patients, particularly between LGE(+) and LGE(-) groups. Specifically, in patients with DCM, the abnormal elevation of gal-3 level indicated a greater risk of MACEs. Moreover, the subgroup analysis according to the study design showed that the above result was statistically significant in prospective studies, but not in retrospective studies. In addition, compared with LGE(-) group, the level of gal-3 in LGE(+) group was higher, which indicated the combined use of gal-3 and LGE significantly improved the prediction of MACEs.

As we all know, gal-3 had been previously proved to be a prognostic biomarker of various heart diseases, such as acute or chronic heart failure (*de Boer et al., 2011*; *Lok et al., 2013*; *van Vark et al., 2017*), valvular heart disease (*Kortekaas et al., 2013*), even in patients with heart failure (*Gopal et al., 2012*), the level of gal-3 is negatively correlated with renal function. Based on a large number of *in vitro* studies, galectin-3 had been identified as an important fibrogenic protein (*Calvier et al., 2015*), and *in vivo* studies had also proved that this fibrogenic effect of gal-3 may be closely related to the prognosis of heart failure. For example, the expression of gal-3 was markedly increased in rats that later developed into heart failure(*Sharma et al., 2004*). Moreover, disrupting gal-3 genetically and inhibiting

its levels pharmacologically mitigated cardiac fibrosis, left ventricular dysfunction, and ensuing heart failure in murine models (*Yu et al., 2013*). Therefore, the study evaluated the level of gal-3 in patients with DCM and found that the abnormal elevation of gal-3 level indicated a greater risk of MACEs, which proved that gal-3 had significant predictive value for MACEs in DCM. Additionally,the reason why the result was statistically significant in prospective studies, but not in retrospective studies may be the limitation of the number of included studies.

Cardiac MRI with LGE is a non-invasive technique that precisely delineates MF and infiltration areas (*Masci et al., 2012*) in DCM, representing the most effective current method for evaluating MF. Previous studies had shown that the existence and degree of LGE in patients with non-ischemic dilated cardiomyopathy (NIDCM) were independently related to heart failure, malignant ventricular arrhythmia, cardiac death and all-cause mortality (*Gulati et al., 2013*; *Halliday et al., 2019*; *Weir et al., 2013*). However, LGE still has some limitations in detecting myocardial scar, and not every DCM patient will perform LGE (*Karatolios et al., 2018*) (such as claustrophobia, obese patients and metal implants). Therefore, some studies suggested that confirming CMR by combining some serum biomarkers may be helpful for stratifying risks and improving the accuracy of diagnosis (*Cojan-Minzat, Zlibut & Agoston-Coldea, 2021*). Among all serum biomarkers, gal-3 has the strongest predictive ability, and it has been proved that combined with LGE,gal-3 is still independent predictors, even after adjusting standard covariates such as age, sex, renal function and NT-pro BNP (*Revnic et al., 2022*). Therefore, the study evaluated the expression of gal-3 in LGE(+) and LGE(-) groups, and found that the level of gal-3 in LGE(+) group was higher than that in LGE(-) group, which indicated gal-3 may be related to myocardial fibrosis in DCM patients in LGE(+) group, and indicated that the combination of gal-3 and LGE can significantly enhance the prediction of MACEs. In addition, it should be noted that serum gal-3 reflects the systemic metabolism of collagen, not just cardiac collagen, which may explain to some extent why these serum markers can be accurately evaluated only while they are jointly predicted with the cardiovascular imaging parameter LGE (*González et al., 2018*). In addition, it has been suggested that in the future, the combination gal-3 and LGE (*Ferreira et al., 2019*; *Xu et al., 2021*) may deserve further study in detecting the disease progress and determining which patients will benefit from implantable cardioverter or cardiac resynchronization therapy.

Considering the limitations of this meta-analysis is crucial for interpreting its findings. First, a potential language bias may arise from the inclusion of only English-language articles. Second, the outcome might be affected by various factors, including the character of the study population (age, gender, gal-3 cut-off value). Particular attention must be given to the range of gal-3 cut-off values identified across the studies, which spanned from 4.1 ng/ml to 59 ng/ml. Such variability is far from inconsequential, given the critical role of gal-3 levels in elucidating the pathophysiological underpinnings of the conditions under study. Such heterogeneity could lead to the over or underrepresentation of actual effect sizes. Also, the inconsistencies in age, gender, baseline cardiac function status, and disease duration among patients across the studies existed. These variations could introduce an additional layer of complexity and potential bias to our analysis. However, due to

the limited sample size and information in each study, the study cannot perform more subgroup analyses. Therefore, we emphasize the importance of adopting standardized methodological approaches in future studies to enhance the consistency and reliability of findings. Thirdly, the number of literature included is limited. The discussion on gal-3 level in LGE(+) *vs* LGE (-) group is of great clinical significance, but the number of studies that can be included is very limited, which may be the main reason for the publication bias. Then the study performed the trim and fill analysis and found that the results were stable.

## CONCLUSION

The results of this study suggest that abnormally elevated gal-3 levels are associated with a significant increase in MACE of DCM and may be a rather potential biomarker. However, there is significant heterogeneity among existing studies, and more studies are needed to determine the predictive value of gal-3 in the prognosis of DCM, especially the diagnostic value of gal-3 combined with LGE.

### Funding
The authors received no funding for this work.

### Competing Interests
The authors declare there are no competing interests.

### Author Contributions
- Yan Xiong performed the experiments, analyzed the data, prepared figures and/or tables, and approved the final draft.
- Qing Zhang conceived and designed the experiments, authored or reviewed drafts of the article, and approved the final draft.

### Data Availability
   The raw measurements are available in the Supplementary Files.

### Supplemental Information
Supplemental information for this article can be found online at http://dx.doi.org/10.7717/peerj.17201#supplemental-information.

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
