# Peer review of "Prognosis value of galectin-3 in patients with dilated cardiomyopathy: a meta-analysis"

_PeerJ, doi:10.7717/peerj.17201_

## Round 0.1 · original submission · Major Revisions

Dear Dr. Zhang,

The manuscript deals with the prognostic marker for DCM, which is a very interesting topic, and shows a very clear result that glectin3 could act as a potential marker for the DCM prognosis. Although the reviewers agree largely with your point, some concerns have been raised. Please try to answer all the reviewers' comments, especially that from reviewer 2. Thanks.

Best,
Jian Song

·

Basic reporting

This paper presents a meta-analysis that evaluates the prognostic value of galectin-3 (gal-3) in patients with dilated cardiomyopathy (DCM). The study includes a comprehensive collection of seven studies and analyzes the association between gal-3 levels and major adverse cardiovascular events (MACEs) in patients with DCM. The results show that elevated gal-3 levels indicate a greater risk of MACEs in DCM patients.Additionally, the study examines the combination of gal-3 and late gadolinium enhancement (LGE) as a predictor of MACEs. The findings suggest that the combination of gal-3 and LGE significantly improves the prediction of MACEs in DCM patients. Overall, the manuscript draws a positive conclusion.

Experimental design

This meta-analysis has a complete structure, rigorous design, and certain clinical significance with a technical and ethical standard. However, the study does not provide detailed information on the methodology of the included studies, such as the diagnostic criteria used for DCM or the specific gal-3 cut-off values.

Validity of the findings

The study makes a positive conclusion from a relatively small amount of research. Have you considered potential confounding factors, such as age, gender, and comorbidities, in the analysis? Conclusion needs to be revised, the current results do not support a "significant" prognostic factor. I think for most comments authors may not have any data, so the best for them is to modify and moderate the conclusion, justify, and list limitations, with this I think the study can be published.

Reviewer 2 ·

Basic reporting

Clear and unambiguous, professional English used throughout.
Literature references, sufficient field background/context provided.
Professional article structure, figures, tables. Raw data shared.
Self-contained with relevant results to hypotheses.

Experimental design

Original primary research within Aims and Scope of the journal.
Rigorous investigation performed to a high technical & ethical standard.

Validity of the findings

All underlying data have been provided; they are robust, statistically sound, & controlled.

Additional comments

Also see the attached PDF with the following Major concerns:

1. The formatting of the references appears to be incorrect and there is no way to know the
effectiveness of the references cited by the authors.
2. The authors included the seven studies in their analysis derived HR values differently,
with log-rank tests via cut-offs or Cox regressions. This again leads to bias and unreliable
results.
3. The detection of Gal-3 varied from study to study, and I suggest that the authors add a
relevant description in the literature search section.
4. The subgroup analyses provided by the authors were not sufficient, and I would suggest
that the authors conduct new subgroup analyses based on the Gal-3 measurement
modality/treatment modality/statistical methodology of the study.

Annotated reviews are not available for download in order to protect the identity of reviewers who chose to remain anonymous.

Reviewer 3 ·

Basic reporting

The English and scientific terminology used in this article is generally clear and well-structured. The author effectively elucidated the potential role of Gal-3 in predicting DCM.

However, some sentences are lengthy, and breaking them into shorter sentences could enhance readability. Certain phrases such as “is closely related to” “as we all know” could be refined for greater precision and interpretation.

Gal-3 is extensively documented for its association with a wide spectrum of heart diseases, extending beyond DCM. It is necessary to provide a comprehensive background in this context to present a holistic view of the research landscape in this field.

Experimental design

Research topic is clearly stated and consistently addressed throughout the article, focusing on evaluating the predictive value of Gal-3 in DCM diagnosis.

Could the authors provide more details on the search strategy used for each database, especially the rationale behind selecting specific keywords and MeSH terms?

Considering the limitations mentioned for endocardial biopsy and LGE, are there alternative methods or emerging technologies being explored for the diagnosis and risk assessment of DCM?

In many of the referenced studies, the observed fold change is subtle, albeit statistically significant. How do the authors substantiate the reliability of such a minor change in clinical use.

As the authors pointed out in the discussion, the demographic characteristics of this study, such as age, gender, Gal-3 cutoff amount, etc may be tremendously affect the predictive value of Gal-3, thus warranting further evaluation.

In addition to Gal-3, is there any other proteins that may potentially serve as a biomarker of DCM? Use them as a control may help corroborate the conclusion.

Validity of the findings

It seems the correlation between Gal-3 and the occurrence of cardiac diseases has been extensively investigated. How does Gal-3 specifically contribute to the prediction of DCM, compared to its involvement in other cardiac diseases? Is it conceivable that it functions as a general marker of fibrosis or inflammation?

Additional comments

no additional comments

---

## Round 0.2 · accepted · Accept

Hereby confirm I that the authors have addressed all of the reviewers' comments.

The Section Editor noted: "The abstract must be carefully edited to eliminate any grammar or textual errors and define abbreviations at their first mention."

·

Basic reporting

The author responded and resolved all the issues raised by the reviewers. I suggest accepting this article.

Experimental design

No other problems.

Validity of the findings

No other problems.

Additional comments

No other problems.

Reviewer 2 ·

Basic reporting

no comment

Experimental design

no comment

Validity of the findings

no comment

Additional comments

The 2nd edition manuscript has been revised better than the 1st edition. And the new manuscript is recommended to be accepted by the journal PeerJ.

Reviewer 3 ·

Basic reporting

The language use is appropriate. Literature references generally cover the research context. The overall structure and layout of the story are logically consistent.

Experimental design

Research question is important and meaningful. Methods are elucidated clearly and sufficiently. Overally, this work provided insight into an new predictive apporach for assisting DCM diagnosis.

Validity of the findings

Looks data processing and statistics were performed in rigorous manner. Conclusions are also clealy claimed.